# UAV Fleet as a Dependable Service for Smart Cities: Model-Based Assessment and Application

**Vyacheslav Kharchenko** [1] , **Ihor Kliushnikov** [1] , **Andrzej Rucinski** [2] , **Herman Fesenko** [1] and **Oleg Illiashenko** [1,*]

1   Department of Computer Systems, Networks and Cybersecurity, National Aerospace University "KhAI", 17, Chkalov Str., 61070 Kharkiv, Ukraine

2   Department of Electrical and Computer Engineering, University of New Hampshire, Durham, NH 03824, USA

*   Correspondence: o.illiashenko@khai.edu

**Abstract:** The paper suggests a model-based approach to assessment and choice of parameters of unmanned aerial vehicle (UAV) fleets applied as one of the main services for Smart Cities and recommendations to assure their dependability. The principles of building and modeling a UAV Fleet as a Dependable Service (UAVFaaDS) for Smart Cities are formulated. Dependability issues for UAVFaaDS including a taxonomy of UAVF failures caused by equipment faults and attacks on assets were specified. The main results cover methodology, classification of UAVFaaDS models as models of queuing systems, and a set of queueing theory-based models for assessment of UAVFaaDS performance, and availability allowing for analysis and choice of fleet parameters. The efficiency of UAVFaaDS is assessed by the probability of successful delivery of services. The proposed modeling base and algorithms provide a choice of appropriate models for analysis and synthesis of UAVFaaDS, grounding of parameters of UAV fleets considering operation modes, and maintenance policy. The application of the developed models and algorithms during the synthesis of UAVFaaDS allows choosing the appropriate parameters of the fleet and ensuring the dependability of services, as well as service of orders with a probability of 0.9–0.99 depending on the requirements. Two cases of UAVFaaDS application for delivery of medicines in normal and emergence modes, models' development, and recommendations for their utilization are discussed.

**Keywords:** UAV; dependability; smart city; queuing systems; delivery service

## 1. Introduction

### 1.1. Motivation

Conceptions of smart buildings, smart cities, smart regions, and smart ecosystems become more and more hot topics for researchers, engineers, and managers considering the natural desires of people to live in more comfortable, cost-effective, and safe conditions. The most important feature of a smart city that distinguishes it from a "non-smart" one is a set of services provided to residents using modern information, transport, energy, and other technologies. Such technologies are based on special solutions, especially means of end-to-end automation, artificial intelligence, the Internet of Things, etc. [1,2]. These solutions must be integrated as much as possible into everyday work and life, comprehensive, "invisible", and convenient.

Thus, a smart city is a system of smart systems and smart services. Smart systems carry out functions of sensing, analysis, decision making, and actuation in a smart (predictive, adaptive, and efficient) manner to provide human and city community-centric services. There are many services related to health and nutrition, transportation and communication, education and entertainment, security and environment, etc. [3,4]. Services are, so to speak, an initial product that is provided in various states to residents of a smart city or region.

In recent years, the implementation of various services has been carried out not only with the use of information technology (IT), which is their mandatory infrastructure component, but also mobile technology (MT).

The symbiosis of IT and MT creates synergetic e-mobility based on the application of unmanned vehicles, autonomous transport systems, and so on [5]. Unmanned aerial vehicles (UAVs) become the leading technology for the deployment of smart city services (SCSs) [6–8]. UAVs can provide the majority of SCSs connected with the delivery of food, goods, medicines, and so on, monitoring and alarming security cases, entertainment and advertising services, and others. The emergence of the concept of UAV as a service (UAVaaS) was a natural step in expanding the use of UAVs for SCSs [6,9].

Development of the concept of UAVaaS requires a transition to the more efficient and systematic use of UAVs in smart cities considering many circumstances and tendencies such as [10–13]:

- Increasing diversity and intensiveness of their application, the necessity of implementation of different types of UAVs and UAV-based groups (swarms, flocks, fleets) to solve dynamical tasks for smart cities and critical infrastructures;
- The complication of the conditions for the use of drones and the growing influence of the physical and cyber information environment on the performance of tasks by drones;
- Implementation of artificial intelligence (AI) for supporting key functions of UAVs' application (navigation, image recognition, the safety of flights, control and optimization of joint behavior of drones, and so on);
- The importance of establishing centralized maintenance, ensuring the reliability and availability of UAVs;
- The need to ensure a more efficient UAV application (reducing costs while maximizing reliability, minimizing time indicators, etc.) by the development of mathematical models and methods.
- Thus, the main challenges of UAVaaS improvement are:
- The systematic analysis of its development as a smart city service, the creation of a model basis for research, and enhancing application options considering the parameters of service requests, means of service, and complex environmental conditions;
- Providing dependability (reliability, availability, safety, and security) of UAVaaS in case of extending functions performed and UAV failures caused by different reasons, and trustworthy assessment of service indicators.

### 1.2. State-of-the-Art

The analyzed papers that are devoted to UAV applications for smart cities can be subdivided into three groups: the papers related to general issues of UAV applications including different city services [1,2,6–9,13–24], the papers related to dependability (reliability) issues of UAV applications [25–33], and the papers related to issues on queueing theory utilization for describing UAV functioning [34–42].

*The papers related to general issues of UAV applications.* The UAV-based networking characteristics and protocols, as well as requirements of smart city applications to support the various data traffic flows that are needed between the different networking components, were identified in [1]. Reviewing the research literature on IoT-enabled smart cities, the authors of [2] highlighted the main trends and open challenges related to the adaptation of IoT technologies for the development of sustainable and efficient smart cities. Differential frameworks with heterogeneous smart UAVs for future smart cities were proposed in [6]. The proposed frameworks utilize the existing public transportation including public buses, city trains, and their routes to provide time-sensitive surveillance. Focusing on the measurement of the smartness of smart cities, such as environmental aspects, life quality, public safety, and disaster management, the study of [7] showed approaches aimed at utilizing collaborative drones and IoT to improve the smartness of smart cities based on data

collection, privacy and security, public safety, disaster management, energy consumption, and quality of life in smart cities.

The potential applications integrating UAVs in smart cities (traffic management, environmental monitoring, civil security control, and merchandise delivery) in the context of the technical and non-technical issues were reviewed in [8]. Regulations and enabling technologies available to support such integration were discussed in this study, too. One paper [9] addressed UAV as a Service (UAVaaS) as a cloud orchestration framework aimed at providing efficient coordination and cooperation of commercial UAVs. To perform analysis and testing for UAVaaS integration, this work proposed a simulated environment using off-the-shelf frameworks, web services, and messaging APIs. To tackle the issue of obtaining the physical location information of the data during the data collection in the IoT system, a Low-Cost Physical Locations Discovery (LCPLD) Scheme was proposed in [13]. In the LCPLD scheme, mobile vehicles and UAVs should be utilized for physical location discovery on wireless sensor networks. To detect traffic jams in smart cities by continuously capturing images from different locations, a multi UAV-based solution was proposed in [14].

Features of planning the path of UAVs assigned to fly over the city and check critical points in a smart city were considered in detail in this paper. One study [15] proposed a drone-based IoT as a service (IoTaaS) framework to enable the dynamic provisioning or deployment of IoT devices utilizing drones. Focusing on the implementation of IoTaaS for smart agriculture and air pollution monitoring applications, this study demonstrated the possibility of drone-based IoTaaS in reducing setup costs and increasing the usage of IoT devices. The work of [16] presented an intelligent, autonomous UAV-enabled solution for a traffic monitoring and emergency response handling system utilized in a smart city. The system allowed detection of emergency traffic situations in real-time and investigation of situations for possible accidents with rapid response units via UAV networks. The 5G IoT network formed by a fleet of UAVs for future smart cities was explored in [17]. The proposed network can be used for intelligent transportation systems, automatic industrial systems, smart healthcare, and more.

One study [18] was devoted to the development of a UAV–assisted vehicular ad hoc network (VANET) communication architecture. In this architecture, UAVs fly over the deployed area and provide communication services to the underlying coverage area. The UAV–assisted VANET avails the advantages of line-of-sight communication, load balancing, flexible, and cost-effective deployment. One paper [19] presented a methodology for ensuring a high level of reliability of UAVs utilized in rescue and fire-fighting missions. The results obtained allowed the development of technical recommendations and recommendations on increasing the reliability and performance of UAVs performing rescue and fire-fighting missions. Considering the different tasks performed by drones in smart cities, the authors of [20] focused on Wi-Fi security, drone networking security, and malicious software. It was also noted in this paper that most cybersecurity vulnerabilities are based on sensors, communication links, and privacy via photos.

A comprehensive survey of research aimed at exploring applications of AI in UAV-enabled IoT systems in the smart city environment was conducted in [21]. This study focused on Artificial Intelligence utilization in realizing autonomous and intelligent flying IoT for smart cities. The authors of [22] focus on the utilization of surveillance drones for transportation, environment, infrastructure, object or people detection, disaster management, and data collection. It was shown that a single or multiple UAVs can be applied either as a stand-alone technology or integrated with other technologies (e.g., Internet of things, wireless sensor networks, convolutional neural networks, artificial intelligence, machine learning, computer vision, cloud computing, web applications). A UAV-based long-range environment monitoring system for smart city infrastructure was presented in [23]. The main task of the proposed system was to record the sensory data and visuals of the various gases ($NO_2$, VOC, $CO_2$, $SO_2$) of landfill sites via IoT and UAV. A comprehensive review of UAV edge computing technology for 6G smart environments was provided in [24].

Highlighting the utility of UAV computing, this study noted the critical role of Federated Learning in meeting the challenges related to energy, security, task offloading, and latency of IoT data in smart environments. This paper also noted that UAV and IoT energy limitations, wireless communication security, data privacy, and scalability of Machine Learning solutions are to be paid more emphasis in the domain of UAV-assisted MEC.

*The papers related to dependability (reliability) issues of UAV applications.* One paper [25] demonstrated features of the reliability assessment for a UAV system both at the specific architecture selection stage for optimizing the system design and at the design stage for avoiding any critical failure that may result in a catastrophic effect on the UAV. To establish a more efficient interval for the maintenance activities for UAVs, a new logistic approach based on reliability and maintenance assessment was presented in [26]. The approach allowed the development of the concepts of preventive and corrective maintenance that consider the system subjected to partial performance degradation. To support the dependability analysis/modeling of UAVs, a model-based approach was proposed in [27].

The approach allowed reducing errors in performance dependability analysis through the automatic synthesis of dependability artefacts required for certification. One paper [28] discussed reliability models for a multi-fleet of drones with one- and two-level systems of control stations and formulated recommendations for the choice of structures of such systems. Reliability analysis of a drone fleet comprising homogenous and heterogeneous components was performed in [29]. The reliability index in this paper was calculated via the structure function.

One work [30] was devoted to the utilization of consistent monitoring, control, and management scheme to guarantee the reliability and performance of the services of a programmable drone fleet. This scheme is a part of the presented strategy aimed at providing the number of drones needed to meet certain levels of service availability. In [31], stochastic Petri net and reliability models were utilized to assess the availability and reliability of the UAV-based IoT system where the UAVs were used to route data from IoT devices to the edge/cloud through a base station. To maximize the mission reliability of a UAV swarm by reasonably arranging the number of different UAVs in the swarm, an optimization model was developed in [32]. The calculation method of mission reliability of the UAV swam was given by modeling the swarm mission as a k-out-of-n system with phased requirements. In [33], the authors developed a new reliability modeling approach called SafeDrones aimed at enabling runtime reliability and risk assessment of UAVs. The approach allowed UAVs to update their missions in an adaptive manner.

*The papers related to issues on queueing theory utilization for describing UAV functioning.* When developing a novel multi-channel load awareness-based MAC protocol for the flying ad hoc network, the multi-priority queueing and service mechanism was proposed in [34]. This mechanism was modeled via the multi-class queueing theory and its implementation allowed efficiently utilizing the network bandwidth resource. To support the computation-intensive and latency-critical services for a UAV swarm, the enhanced collaborative computing model based on queueing theory was developed in [35]. The model allowed achieving the closed-form solution of the decision threshold.

One study [36] demonstrated how, under certain assumptions, a military intelligence unit equipped with multiple identical UAVs can be treated using a queueing theory methodology and/or as Markov chains. Using these models, the steady-state probabilities of the unit, its performance effectiveness, and various performance parameters were calculated. To account for the waiting times of drones receiving battery swap services at automated battery maintenance system (ABMS) locations, an optimization model incorporating a multiclass M/D/1 queueing model was developed in the work of [37]. The proposed model also allowed for deciding optimal locations for ABMSs for a given candidate set. To explore the influence of each of the considered charging stations' limited capacity and spatial density on the performance of a drone-enabled wireless network, special tools from queuing theory and stochastic geometry were utilized in the paper of [38].

To calculate the fleet size allowing the given part of the fleet to stay in the air perpetually, a rule-of-thumb formula based on Queuing Theory and Energy Conversation was derived in the study of [39]. The paper of [40] considered UAVs utilized to build flying ubiquitous sensor networks as a queuing system and their swarm—as a queuing network. Modeling results showed that a relatively small number of UAVs to estimate the time of the data delivery for a relatively small number of UAV familiar approximate estimates for systems G/G/1 could be applied. In [41], the authors modeled drones utilized as aerial content delivery points in the Mobile Content Delivery Network (CDN) architecture and base station for them via queuing theoretical models. This modeling allowed obtaining blocking probabilities with the Erlang-B parameter to determine additional drone transfer. The study of [42] suggested a model of a UAV-based system for monitoring NPP accidents by use of queueing theory and reliability models considering routings covered by drones during monitoring activities. At the same time, aspects of a more detailed assessment of UAV fleet efficiency considering failures and recovery procedures were also addressed.

### 1.3. Aim and Objectives

The main conclusions based on the analysis of publications are that UAVs and UAV groups (swarms, fleets, and so on) are not considered dependable systems for providing services in a smart city considering various requests. The study aims to develop a model-based approach to the assessment and choice of parameters of UAV fleets applied as one of the main services for smart cities and recommendations to assure the dependability of this service.

The objectives of the research are:

- To formulate the principles of building, modeling, and implementing a UAV Fleet as a Dependable Service (UAVFaaDS) for a Smart City;
- To specify dependability issues (analysis and assessment) for UAVFaaS for a Smart City including a taxonomy of UAV and UAVF failures caused by equipment faults and attacks on assets;
- To develop and explore queueing-theory-based models of UAVFaaDS for a Smart City;
- To describe and analyze two cases of UAVFaaDS for a Smart City.

The paper is structured as follows. The next Section 2 grounds the principles of the building of a UAV Fleet as a Dependable Service for a Smart City (Section 2.1) and describes the main dependability aspects of UAVFaaDS (Section 2.2). Section 3 proposes the results of developing and exploring queueing-theory-based UAVF models. Section 4 describes two cases of application of UAVFaaDS for delivery of medicines in peace and extremal conditions, results of models' analysis, and recommendations for their utilization. Recommendations for using the developed models are presented in Section 5. Section 6 discusses the results of the research. The last Section 7 provides conclusions and describes areas for further research.

The main contribution of the research includes methodology and a set of models for assessment of UAVFaaDS performance, availability, and maintainability characteristics allowing for analysis and ground choice of UAV fleet parameters for its application as an SCS. The novelty of the work is in the use of queueing-theory-based models and Markov chains for the analysis of existing UAV services and the synthesis of UAVFaaDS with required performance and dependability characteristics.

## 2. Approach and Methodology

### 2.1. Concepts of UAVF as a Dependable Service

The proposed approach is based on the concept of UAVFaaDS and the corresponding principles of its presentation and implementation. The concept of "UAV fleet as a reliable service" consists of the formation and application of a set of multifunctional UAVs to provide specified services for a smart city, the performance of which is ensured by systems of control, continuous maintenance, replenishment, and recovery in case of failures caused by physical and informational influences.

The key principles and components of the UAVFaaDS concept are the following:

(a)    A UAV fleet is a set of different UAVs, which is described by formula:

$$SFleet_{UAV} = \{SSwarm_{UAV}, SFlock_{UAV}, SSep_{UAV}, IS_C, IS_M\}, \qquad (1)$$

where

$SSwarm_{UAV}$—a set of swarms $\{SSwarm_{UAVi}, i = 1,2, \ldots, w\}$. A swarm is a set of UAVs applied for a joint goal, i.e., performing some service, w—number of swarms;

$S_{UAVS}$—a set of flocks $\{SFlock_{UAVj}, j = 1,2, \ldots, f\}$. A flock is a set of UAVs applied for a few different goals, i.e., performing a few different services, f—number of flocks;

$SSep_{UAV}$—a set of separate UAVs $\{Sep_{UAVk}\}$ of different types, which are applied to provide the performance of different services or redundancy for swarms and flocks;

$IS_C$—a system for UAVF control. It consists of a set of control stations $\{CS_c, c = 1,2, \ldots, s\}$ to provide control for a UAV fleet and its components;

$IS_M$—a system for UAVF maintenance. It consists of a set of automation battery systems $\{AS_b, b = 1, 2, \ldots, a\}$ and other means for recovery and maintenance.

Additionally, the UAV fleet is a natural and obvious component, mobile and smart subsystem of the Smart City/region ecosystem.

(b)    A UAV fleet provides a set of services $\{SSerq, q = 1, \ldots, e\}$, which are a subset of the main services of a smart city, and is a part of the general city UAV fleet (GUAVF) formed for providing of targeted services.

(c)    Considering that the majority of functions and services are sensitive in point of view dependability issues, the important part of the concept is assurance of specified system dependability and resilience attributes [43,44]. Thus, UAVF should be presented as a dependable multi-service system.

### 2.2. Dependability of UAV Fleet

In the context of the SCSs and considering [43], the dependability of UAVF and its subsystems can be defined as the ability to deliver required services for smart cities that can justifiably be trusted. The main attributes of dependability, which are important for UAVF, are the following:

- Reliability is defined by continuity of correct service(s) delivered by drones or UAVF during the required time;
- Availability is described as the readiness of drones or the UAVF to deliver correct service(s) at any time;
- Maintainability describes the ability of drones or the UAVF to undergo modifications, recharges, and repairs considering time limitations for delivery of services;
- Safety is defined by the absence of negative/critical consequences on the customers, the environment, and other systems in process of service delivery by drones or UAVF;
- Security is described by a set of attributes of integrity, confidentiality, and accessibility. Integrity is a key attribute and is defined by the absence of improper alterations for on-board drone systems and IT infrastructure of UAVF (systems ISC, ISM). Confidentiality and accessibility are abilities to protect the cyber assets of UAVF against unauthorized access and to assure access for authorized customers.

To assess the dependability of UAVF for SCSs it is required to analyze the taxonomy of failures caused by different faults [44]. A set of different UAV faults can be classified according to a number of features (reasons, stages of manifestation, consequences, and so on. There are three groups of failures and faults, which should be considered and analyzed:

- Physical faults, which lead to persistent or short-term failures of drones and station hardware;
- Design faults caused by erroneous actions during the specifying system and software requirements, development, and verification. Such faults are the most typical reasons for failures for software or firmware of drones and UAV IT infrastructure;

- Interaction faults and failures caused by them, which are the result of the influence of external physical influences and cyber intrusions/attacks on drones and UAVF hardware and software vulnerabilities.

To assure dependability, the following are applied:

- Structure redundancy of UAV fleets by use of additional drones, stations, and other system components [12,28];
- IMECA-based analysis of Internet of Drones vulnerabilities and choice of countermeasures to minimize risks of successful cyberattacks [45];
- Sensor, communication, and data processing diversity of UAV-based systems [46];
- Strategies of UAVF maintenance including charging, recharging, and recovery after failures caused by different faults and influences.

To develop models of UAVF application considering performance and dependability, it is suggested to apply the mathematical apparatus of queueing theory. More simple models such as reliability models of redundant UAVFs were investigated in many works [25–28]. Cybersecurity aspects can be considered by specifying failure rates of drones and other systems of UAVF. Critical attacks on the IT infrastructure of UAVF can be assessed by the probability of independent events as a general indicator of dependability.

## 3. Queueing-Theory-Based Models of UAVFaaDS

### 3.1. Classification of Queueing-Theory-Based UAVFaaDS Models

In general, models of queueing systems are described by the Kendall–Lee notation:

$$(A/B/C):(D/Q/K), \qquad (2)$$

where A—the distribution of inter-arrival times; B—the distribution of service times; C—the number of servers (C = 1 … n, ∞); D—the queueing discipline (FCFS (first come, first served), LCFS (last come, first served), SIRO (served in random order), SRPT (shortest remaining processing time), LRPT (longest remaining processing time), RR (round-robin), and others); Q—the system capacity (Q = 0 … p, ∞); K—the number of kinds of customers (K = 1 … r).

In this paper, the addressed queueing-theory-based models of UAVFaaDS are as follows:

- Models without loss of customers;
- Models with loss of customers due to UAV failures.

Parameters A and B mentioned above can be designated in the following way: M—Markovian or Exponential distribution; E—Erlang distribution (a special case of the Gamma distribution); D— Deterministic distribution; G/GI—General/General independent distribution; H—hyper exponential distribution.

To provide a classification of queueing-theory-based models of UAVFaaDS, the following aspects are considered:

- UAVs and UAV swarms are considered as the servers;
- Different kinds of customers may require different configurations of servers: one UAV is enough to deliver cargo, but a UAV swarm is needed for monitoring missions, or, if it is necessary, for deploying a flying wireless network;
- Considered queueing systems have unlimited recovery;
- Maintenance of UAVs is conducted during customer servicing;
- Redundant UAVs are used to change the failed ones.

The limited flight time of the UAV requires charging (replacing) the onboard power sources for further operation. To cope with this problem, ABMSs are used.

A faceted classification of queueing-theory-based models of UAVFaaDS that are considered in this work is shown in Figure 1.

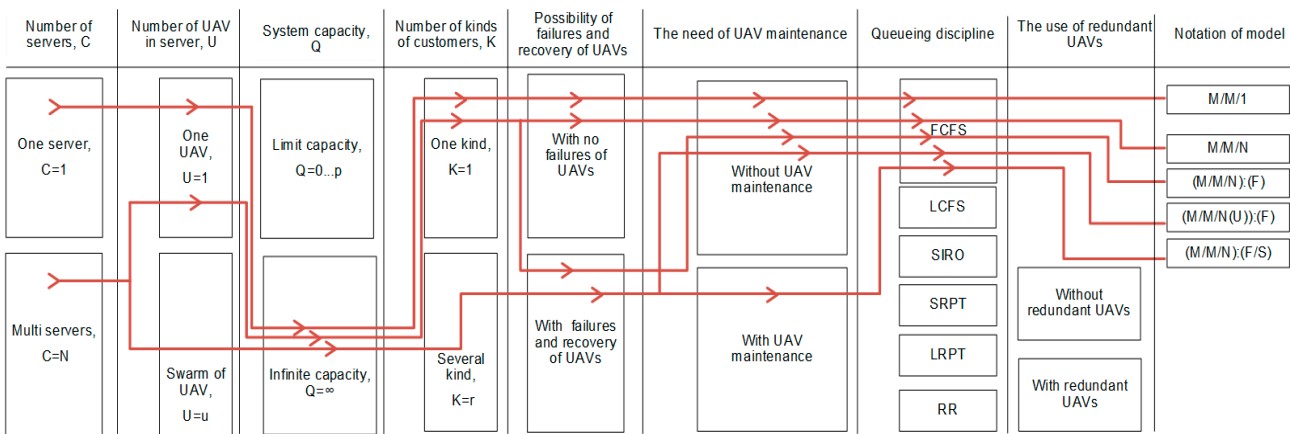

**Figure 1.** Faceted classification of queueing-theory-based models of UAVFaaDS.

Based on the presented faceted classification, Equation (1) can be rewritten in the following way:

$$(A/B/C(U)):(D/Q/K):(F/S)/R \tag{3}$$

where U—the number of UAVs in one separate server (U = 1 ... u); F—the model providing for the possibility of UAV failures and their recovery; S—the model providing for UAV maintenance during customer servicing; R—the model providing for the use of redundant UAVs.

To evaluate the operation of UAVFaaS, the following indicators are used in the work:

The probability of the successful service delivery ($P_{ssd}$), which is determined by the sum of the probabilities that the system is in an operational state to wait and process applications;

The probability of the idle state of the system ($P_{is}$), which is determined by the sum of the probabilities that the system will be in an operational state in the absence of applications.

### 3.2. M/M/1 Model of UAVaaS

The M/M/1 is a model with exponential inter-arrival times, exponential service times, one server, infinite queueing capacity, and FCFS discipline. A single UAV is considered a server in this model. The graph of the Markov chain of this model is shown in Figure 2 where $\lambda$ is the arrival rate and $\mu$ the service rate.

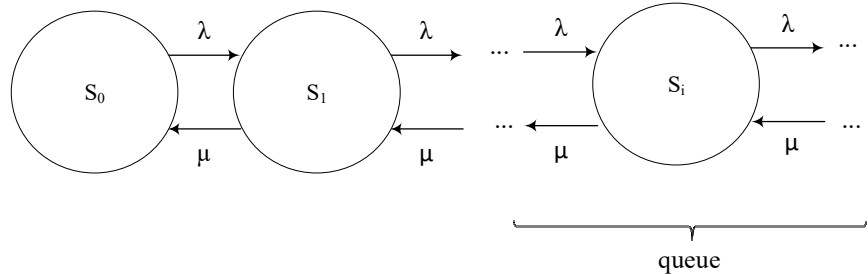

**Figure 2.** Graph of the Markov chain of the M/M/1 model.

The description of the states of the M/M/1 model is given in Table 1.

**Table 1.** Descriptions of the states of the M/M/1 model.

| State | Description of the State |
|---|---|
| $S_0$ | No customers are in the system |
| $S_1$ | The system services one customer |
| $S_i$ | $i$ customers are in the system, one customer is serviced, $(i-1)$ customers are in the queue |

This model can be used to analyze the operation of UAVaaS with only one UAV under the assumptions of its absolute fail-safe and unlimited flight time. The use of this model makes it possible to determine the limit values of order flow parameters and their maintenance and justifying the characteristics of UAVs, which are necessary for the deployment of the dependable service.

### 3.3. M/M/N Model of UAVaaS

The M/M/N is a model with exponential inter-arrival times, exponential service times, *N* servers, infinite queueing capacity, and FCFS discipline. A single UAV is considered as one server and a swarm of UAVs is considered as all servers in this model. The graph of the Markov chain of the M/M/N model is shown in Figure 3.

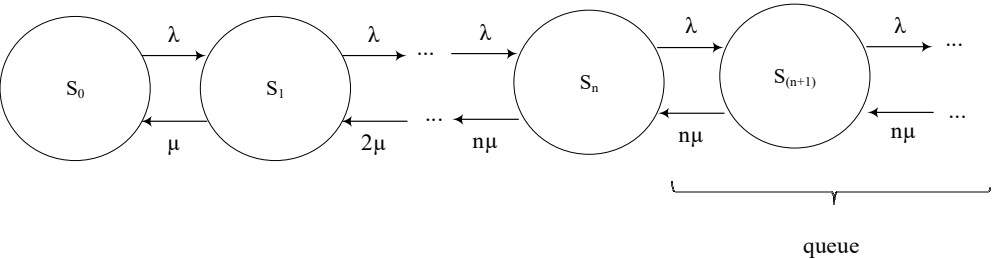

**Figure 3.** Graph of the Markov chain of the M/M/N model.

The descriptions of the states of the M/M/N model are given in Table 2.

**Table 2.** Descriptions of the states of the M/M/N model.

| State | Description of the State |
|---|---|
| $S_0$ | No customers are in the system |
| $S_1$ | The system services one customer |
| $S_n$ | The system services $n$ customers |
| $S_{(n+1)}$ | $(n + 1)$ customers are in the system, $n$ customers are serviced, one customer is in the queue |

This model can be used to analyze the operation of a simple service based on the UAVFaaS fleet with assumptions about absolute failure and unlimited UAV flight time. This model allows determining the value of order flow parameters and their service, as well as justifying the number and characteristics of UAVs necessary for the deployment of the dependable service.

### 3.4. (M/M/N):(F) Model of UAVaaS

The (M/M/N):(F) is a model with exponential inter-arrival times, exponential service times, *N* servers, infinite queueing capacity, and FCFS discipline. A single UAV is considered as one server and a swarm of UAVs is considered as all servers in this model. The UAVs can fail and be restored during the customer servicing. The customer that is being serviced is lost as a result of the UAV failure. The failure of one UAV in the swarm leads to the termination of the operation of the entire swarm and the loss of the customer.

The graph of the Markov chain of the (M/M/N):(F) model is shown in Figure 4 where $\lambda_f$ is the rate of UAV failures and $\mu_r$ the rate of UAV recovery.

In Figure 4, set (I) contains the states describing the operation of the queuing system with up-state UAVs. Set (II) contains states describing the operation of the queuing system with faulty UAVs, which lead to the loss of a customer.

The descriptions of the states of the (M/M/N):(F) model are given in Table 3.

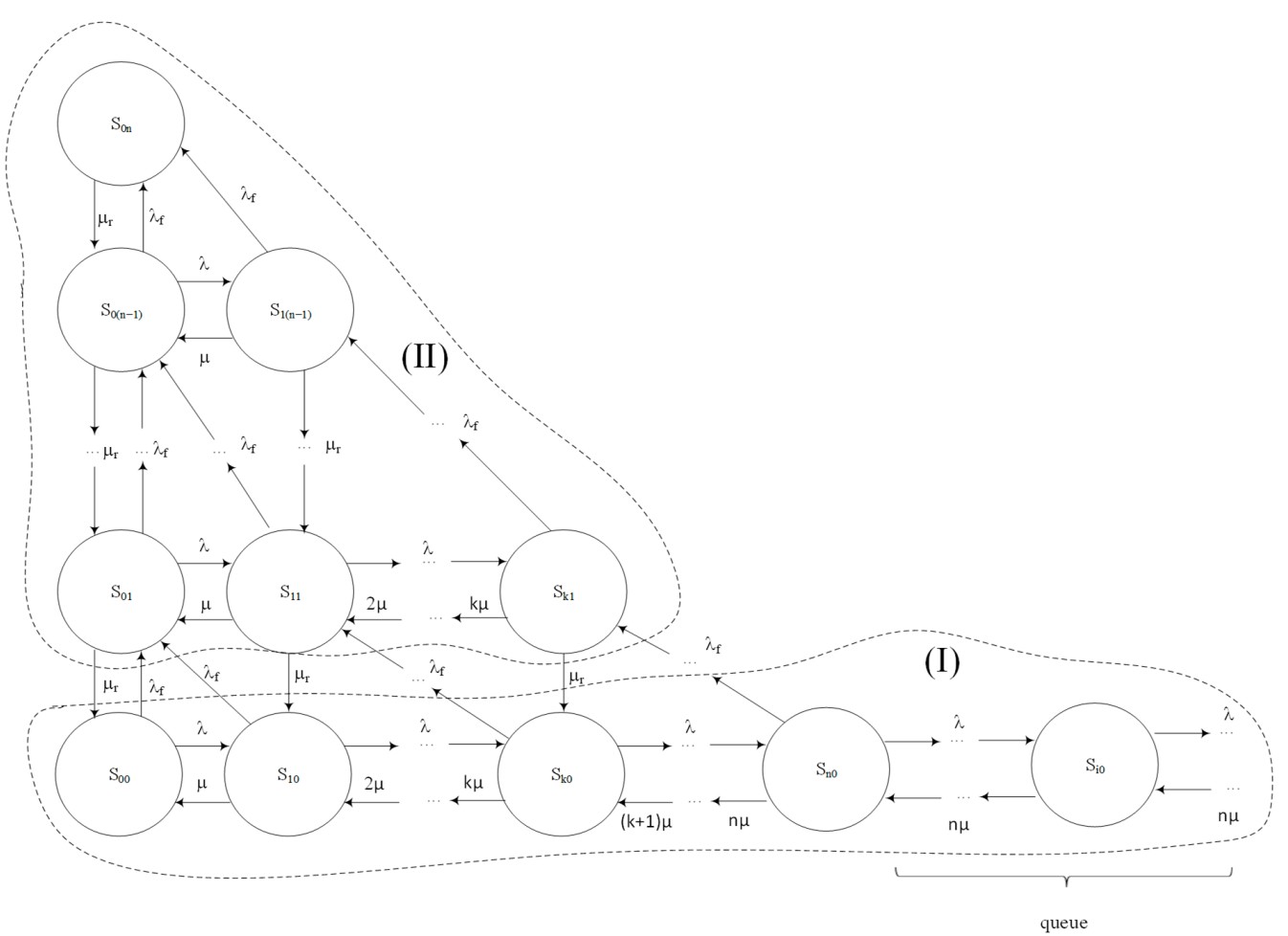

**Figure 4.** Graph of the Markov chain of the (M/M/N):(F) model.

**Table 3.** Descriptions of the states of the (M/M/N):(F) model.

| State | Description of System Status |
|-------|------------------------------|
| $S_{00}$ | No customers are in the system, and all servers (UAVs) are in operable states |
| $S_{10}$ | One customer is in the system, and all servers (UAVs) are in operable states |
| $S_{k0}$ | $k$ customers are in the system, and all servers (UAVs) are in operable states |
| $S_{n0}$ | $n$ customers are in the system, and all servers (UAVs) are in operable states |
| $S_{i0}$ | $i$ customers are in the system, n customers are serviced, all servers (UAVs) are in operable states, and $(i - n)$ customers are in the queue |
| $S_{01}$ | No customers are in the system, and one server (UAVs) is in a non-operable state |
| $S_{11}$ | One customer is in the system, and one server (UAVs) is in a non-operable state |
| $S_{k1}$ | $k$ customers are in the system, and one server (UAVs) is in a non-operable state |
| $S_{21}$ | Two customers are in the system, and one server (UAVs) is in a non-operable state |
| $S_{0(n-1)}$ | No customers are in the system, and $(n - 1)$ servers (UAVs) are in non-operable states |
| $S_{1(n-1)}$ | One customer is in the system, and $(n - 1)$ servers (UAVs) are in non-operable states |
| $S_{0n}$ | No customers are in the system, and all servers ($n$ UAVs) are in non-operable states |

The graph of the Markov chain of the (M/M/3):(-/0/-):(F) model with three UAVs that can fail and be recovered without a queue is shown in Figure 5.

$$
\begin{cases}
-\left(\lambda + \lambda_f\right) P_{00} + \mu_r P_{01} + \mu P_{10} = 0; \\
-\left(\lambda + \lambda_f + \mu\right) P_{10} + \lambda P_{00} + \mu_r P_{11} + 2\mu P_{20} = 0; \\
-\left(\lambda + \lambda_f + 2\mu\right) P_{20} + \lambda P_{10} + \mu_r P_{21} + 3\mu P_{30} = 0; \\
-\left(\lambda_f + 3\mu\right) P_{30} + \lambda P_2 = 0; \\
-\left(\lambda + \lambda_f + \mu + \mu_r\right) P_{11} + \lambda P_{01} + \lambda_f P_{20} + 2\mu P_{21} + \mu_r P_{12} = 0; \\
-\left(\lambda_f + 2\mu + \mu_r\right) P_{21} + \lambda P_{11} + \lambda_f P_{30} = 0; \\
-\left(\lambda + \lambda_f + \mu_r\right) P_{02} + \lambda_f P_{01} + \lambda_f P_{11} + \mu_r P_{03} + \mu P_{12} = 0; \\
-\left(\lambda_f + \mu + \mu_r\right) P_{12} + \lambda P_{02} + \lambda_f P_{11} = 0; \\
-\mu_r P_{03} + \lambda_f P_{02} + \lambda_f P_{12} = 0; \\
P_{00} + P_{10} + P_{20} + P_{30} + P_{01} + P_{11} + P_{21} + P_{02} + P_{12} + P_{03} = 1
\end{cases}
\tag{4}
$$

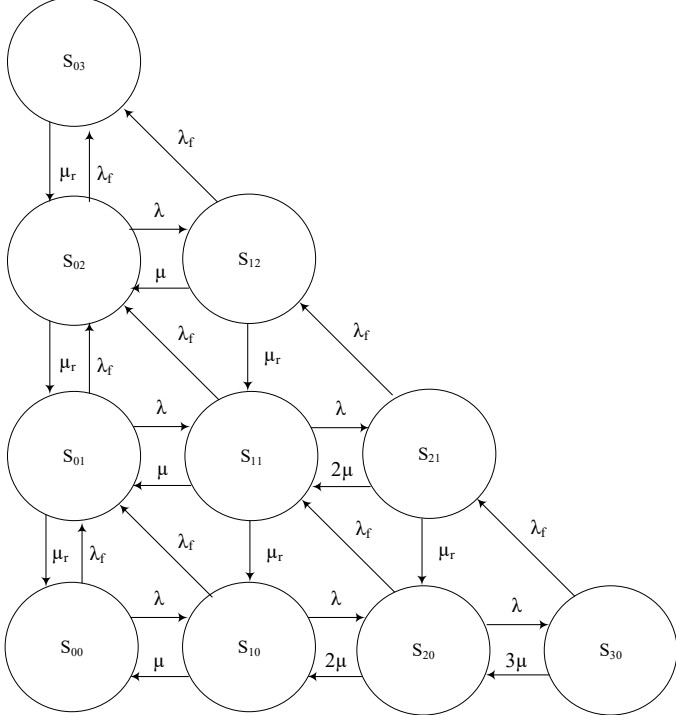

**Figure 5.** Graph of the Markov chain of the (M/M/3):(-/0/-):(F) model.

For $\lambda = 0.5\ 1/h$, $\lambda_f = 0.001\ 1/h$, $\mu = 0.4\ 1/h$, and $\mu_r = 0.5\ 1/h$, the final probabilities of UAVFaaDS being in the given states are as follows: $P_{00} = 0.298$, $P_{10} = 0.372$, $P_{20} = 0.232$, $P_{30} = 0.097$, $P_{01} = 0.95 \cdot 10^{-4}$, $P_{11} = 7.019 \cdot 10^{-4}$, $P_{21} = 3.44 \cdot 10^{-4}$, $P_{02} = 2.527 \cdot 10^{-6}$, $P_{12} = 2.181 \cdot 10^{-6}$, and $P_{03} = 9.416 \cdot 10^{-9}$.

Thus, the probability $P_{ssd} = P_{00} + P_{10} + P_{20} + P_{30} = 0.999$ and the probability $P_{is} = P_{00} = 0.298$.

This model can be used to analyze the functioning of UAVFaaDS, considering the failures of UAVs, which lead to the loss of orders, and with assumptions about their unlimited flight time. This model allows determining the necessary parameters of the service flow and justifying the number and characteristics of UAVs necessary for the deployment of the dependable service.

If UAVFaaDS are described via the (M/M/3):(-/0/-):(F) model, the final probabilities of it being in the given states can be found by solving the following Kolmogorov–Chapman algebraic equations system.

*3.5. (M/M/N[U]):(F) Model*

The (M/M/N[U]):(F) is a model with exponential inter-arrival times, exponential service times, *N* servers with *u* UAVs in each one, infinite queueing capacity, and FCFS discipline. The UAVs can fail and be restored during the customer servicing.

The performance of different types of tasks can be described by different strategies of changing system states due to failures of UAVs in the groups. One type of strategy involves losing the customer that is being serviced, for instance, during deploying flying wireless networks. The behavior of the system with this strategy almost does not differ from the behavior of the system (M/M/N):(F).

In other strategies, UAV failures can only reduce the service rate, for instance, in the case of a search for objects in a defined area by a UAV swarm, when UAV failures lead to an increase in the search time. In this case, the loss of the customer occurs only after the failures of all UAVs in the swarm.

A swarm of UAVs is considered as one server and a fleet of UAVs comprising *N* swarms of UAVs is considered as all servers in this model.

The graph of the Markov chain of the (M/M/N[U]):(F) model is shown in Figure 6 where $\mu_1$ and $\mu_{(u-1)}$ are the service rates of customers in case of failures of one UAV and $(u-1)$ UAVs, respectively.

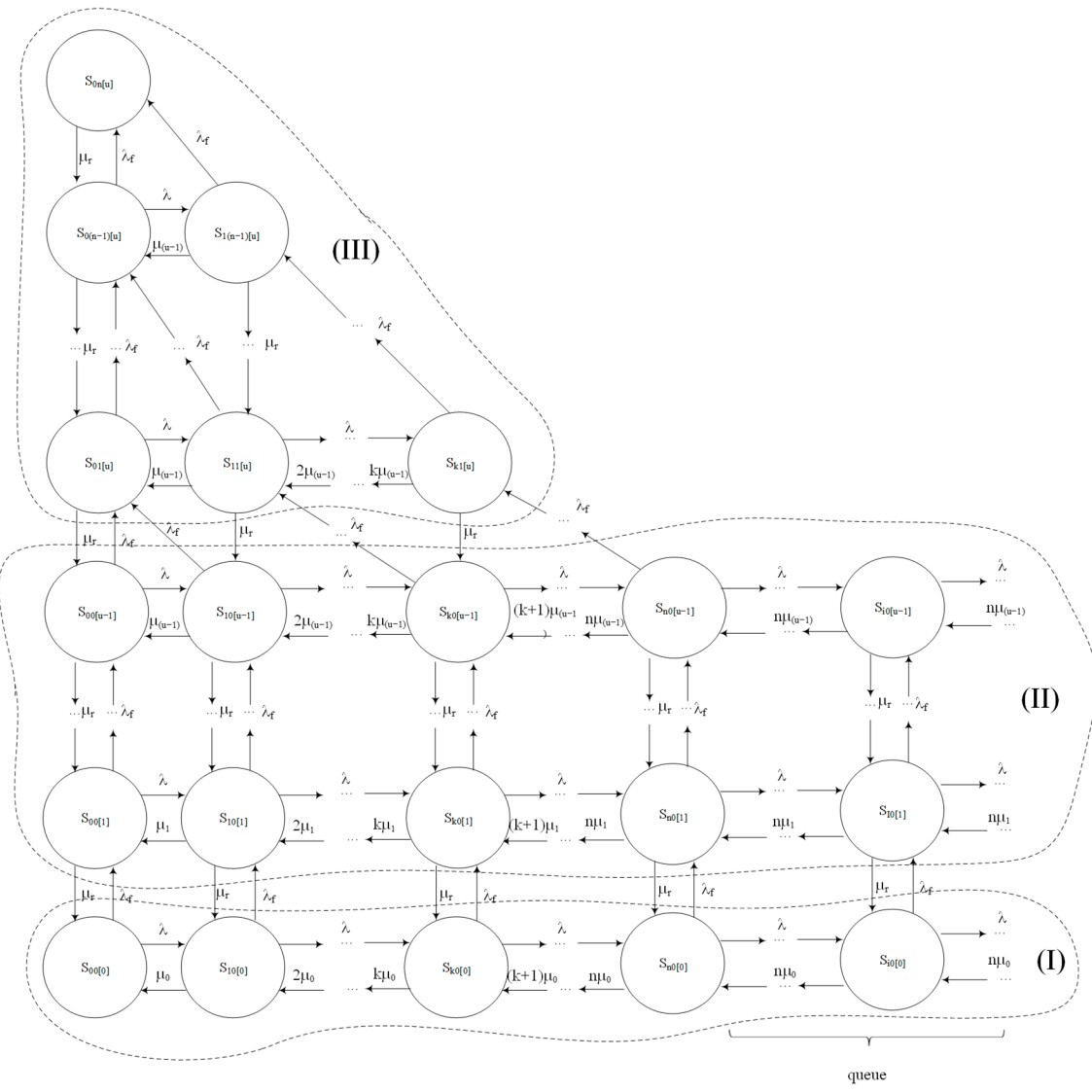

**Figure 6.** Graph of the Markov chain of the (M/M/N[U]):(F) model.

In Figure 6:

- The set (I) contains the states describing the operation of the queuing system (UAVFaaDS) with operable UAVs;
- The set (II) contains states describing the operation of the queuing system (UAVFaaDS) with non-operable UAVs in the swarm for cases when UAVs' faults do not result in the loss of customers;
- The set (III) contains states describing the operation of the queuing system (UAVFaaDS) with non-operable UAVs for cases when UAVs' faults result in the loss of customers.

The descriptions of the states of the (M/M/N[U]):(F) model are given in Table 4.

**Table 4.** The descriptions of the states of the (M/M/N[U]):(F) model.

| State | Description of the State |
|-------|--------------------------|
| $S_{00}[0]$ | No customers are in the system, and all servers (all UAVs in the swarms) are in operable states |
| $S_{10}[0]$ | One customer is in the system, and all servers (all UAVs in the swarms) are in operable states |
| $S_{10}[1]$ | One customer is in the system, and one UAV in one swarm is in a non-operable state |
| $S_{10[u-1]}$ | One customer is in the system, and $[u-1]$ UAVs in one swarm are in non-operable states |
| $S_{k0}[0]$ | $k$ customers are in the system, and all servers (all UAVs in the swarms) are in operable states |
| $S_{k0}[1]$ | $k$ customers are in the system, and one UAV in one swarm is in a non-operable state |
| $S_{k0[u-1]}$ | $k$ customers are in the system, and $[u-1]$ UAVs in one swarm are in non-operable states |
| $S_{n0}[0]$ | $n$ customers are in the system, and all servers (all UAVs in the swarms) are in operable states |
| $S_{n0}[1]$ | $n$ customers are in the system, and one UAV in one swarm is in a non-operable state |
| $S_{n0[u-1]}$ | $n$ customers are in the system, and $[u-1]$ UAVs in one swarm are in non-operable states |
| $S_{i0}[0]$ | $i$ customers are in the system, $(i-n)$ customers are in the queue, and all servers (all UAVs in the swarms) are in operable states |
| $S_{i0}[1]$ | $i$ customers are in the system, $(i-n)$ customers are in the queue, and one UAV in one swarm is in a non-operable state |
| $S_{i0[u-1]}$ | $i$ customers are in the system, $(i-n)$ customers are in the queue, and $[u-1]$ UAVs in one swarm are in a non-operable state |
| $S_{01[u]}$ | No customers are in the system, and one server is in a non-operable state ($u$ UAVs in one swarm are in non-operable states) |
| $S_{11[u]}$ | One customer is in the system, and one server is in a non-operable state ($u$ UAVs in one swarm are in non-operable states) |
| $S_{k1[u]}$ | $k$ customers are in the system, and one server is in a non-operable state ($u$ UAVs in one swarm are in non-operable states) |
| $S_{0(n-1)[u]}$ | No customers are in the system, and $(n-1)$ servers are in non-operable states ($u$ UAVs in $(n-1)$ swarms are in non-operable states) |
| $S_{1(n-1)[u]}$ | One customer is in the system, and $(n-1)$ servers are in non-operable states ($u$ UAVs in $(n-1)$ swarms are in non-operable states) |
| $S_{0n[u]}$ | No customers are in the system, and all $n$ servers are in non-operable states ($u$ UAVs in each swarm are in non-operable states) |

This model can be used to analyze the functioning of UAVFaaDS, considering the failures of UAVs in the swarm, which lead to the loss of orders only after the failures of all UAVs of such a fleet, and with assumptions about their unlimited flight time. The use of this model allows determining the necessary parameters of the service and justifying the number and characteristics of UAVs necessary for the deployment of a dependable service of this type.

### 3.6. (M/M/N):(F/S) Model

The (M/M/N):(F/S) is a model of UAVFaaDS with exponential inter-arrival times, exponential service times, $N$ servers, infinite queueing capacity, and FCFS discipline. A single UAV is considered as one server and a swarm of UAVs is considered as all servers in this model. The UAVs can fail and be restored during the customer servicing. The failure of the UAV results in the loss of the customer served by this UAV. The failure of one UAV in the swarm results in the failure of the entire swarm and the loss of the customers served by this swarm. While customer servicing, maintenance of the UAVs (charging or replacement of on-board power supplies) is conducted as well.

The graph of the Markov chain of the (M/M/N):(F/S) mode is shown in Figure 7 where $\lambda_m$ is the rate of sending the UAVs for maintenance and $\mu_m$ is the rate of UAV maintenance.

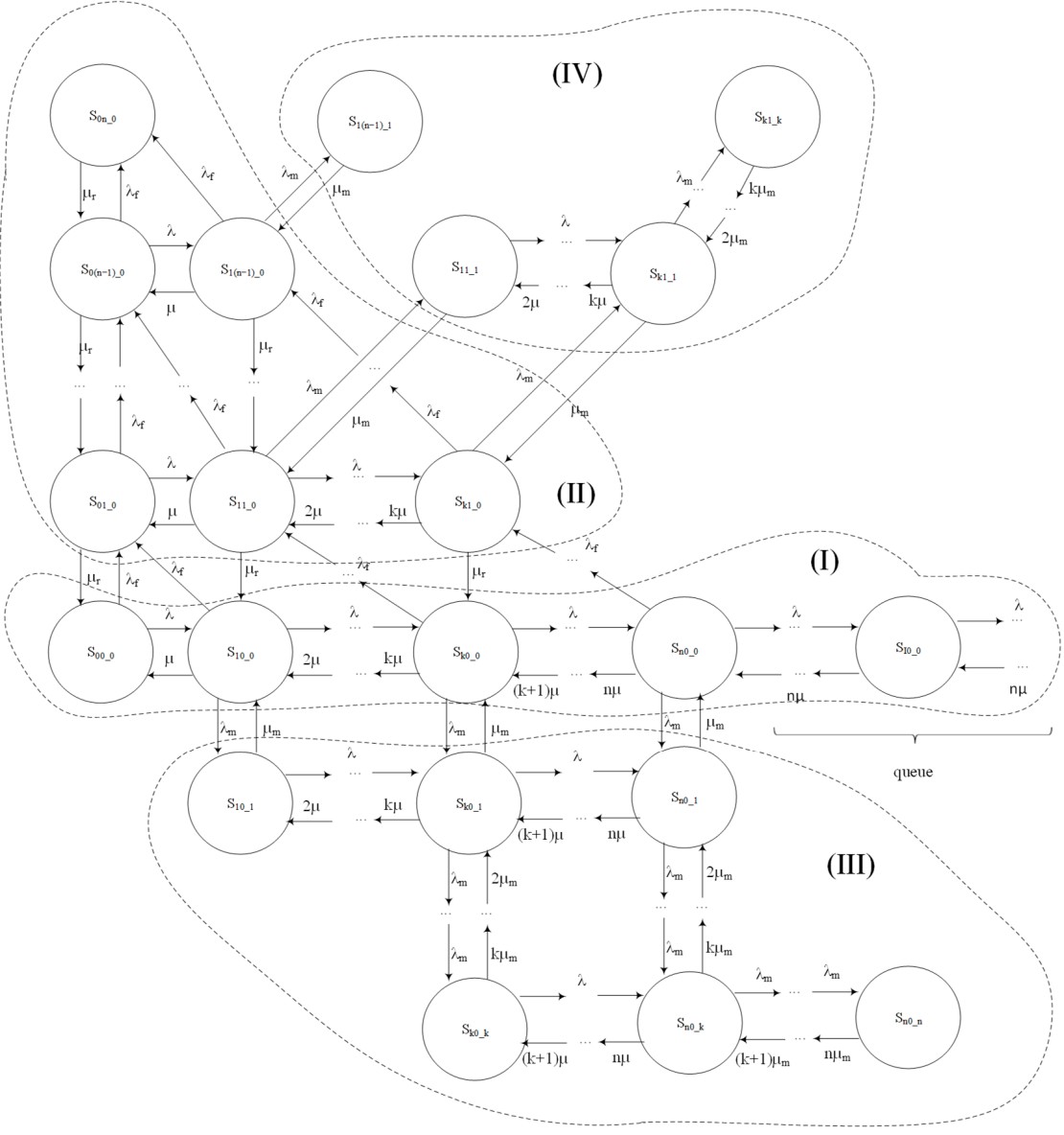

**Figure 7.** Graph of the Markov chain of the (M/M/N):(F/S) model.

In Figure 7:

- The set (I) contains the states describing the operation of the queuing system (UAVFaaDS) with operable UAVs;

- The set (II) contains states describing the operation of the queuing system (UAVFaaDS) with non-operable UAVs;
- The set (III) contains the states describing the queuing system (UAVFaaDS) with the UAVs under maintenance;
- The set (IV) contains states describing the operation of the queuing system (UAVFaaDS) with both non-operable UAVs and UAVs under maintenance.

The descriptions of the states of the (M/M/N):(F/S) model are given in Table 5.

**Table 5.** The descriptions of the states of the (M/M/N):(F/S) model.

| State | Description of the State |
|---|---|
| $S_{00\_0}$ | No customers are in the system, all servers (UAVs) are in operable states, and no servers (UAVs) require maintenance |
| $S_{10\_0}$ | One customer is in the system, all servers (UAVs) are in operable states, and no servers (UAVs) require maintenance |
| $S_{k0\_0}$ | $k$ customers are in the system, all servers (UAVs) are in operable states, and no servers (UAVs) require maintenance |
| $S_{n0\_0}$ | $n$ customers are in the system, all servers (UAVs) are in operable states, and no servers (UAVs) require maintenance |
| $S_{i0\_0}$ | $i$ customers are in the system, $n$ customers are serviced, all servers (UAVs) are in operable states, no servers (UAVs) require maintenance, and $(i - n)$ customers are in the queue. |
| $S_{01\_0}$ | No customer is in the system, one server (UAV) is in a non-operable state, and no servers (UAVs) require maintenance |
| $S_{11\_0}$ | One customer is in the system, one server (UAV) is in a non-operable state, and no servers (UAVs) require maintenance |
| $S_{k1\_0}$ | $k$ customers are in the system, one server (UAV) is in a non-operable state, and no servers (UAVs) require maintenance |
| $S_{0(n-1)\_0}$ | No customer is in the system, $(n - 1)$ servers (UAVs) are in non-operable states, and no servers (UAVs) require maintenance |
| $S_{1(n-1)\_0}$ | One customer is in the system, $(n - 1)$ servers (UAVs) are in non-operable states, no servers (UAVs) require maintenance |
| $S_{0n\_0}$ | No customer is in the system, $n$ servers (UAVs) are in non-operable states, no servers (UAVs) require maintenance |
| $S_{10\_1}$ | One customer is in the system, all servers (UAVs) are in operable states, one server (UAV) is under maintenance |
| $S_{11\_1}$ | One customer is in the system, one server (UAVs) is in a non-operable state, and one server (UAV) is under maintenance |
| $S_{k0\_1}$ | $k$ customers are in the system, all servers (UAVs) are in operable states, and one server (UAV) is under maintenance |
| $S_{k1\_1}$ | $k$ customers are in the system, one server (UAV) is in a non-operable state, and one server (UAV) is under maintenance |
| $S_{k1\_k}$ | $k$ customers are in the system, one server (UAV) is in a non-operable state, and $k$ servers (UAVs) are under maintenance |
| $S_{k0\_k}$ | $k$ customers are in the system, all servers (UAVs) are in operable states, and $k$ servers (UAVs) are under maintenance |
| $S_{n0\_1}$ | $n$ customers are in the system, all servers (UAVs) are in operable states, and one server (UAVs) is under maintenance |
| $S_{n0\_k}$ | $n$ customers are in the system, one server (UAV) is in a non-operable state, and $k$ servers (UAVs) are under maintenance |
| $S_{n0\_n}$ | $n$ customers are in the system, one servers (UAV) is in a non-operable state, and $n$ servers (UAVs) are under maintenance |

Let us consider the (M/M/2):(-/0/-):(F) model, the graph of the Markov chain of which is shown in Figure 8.

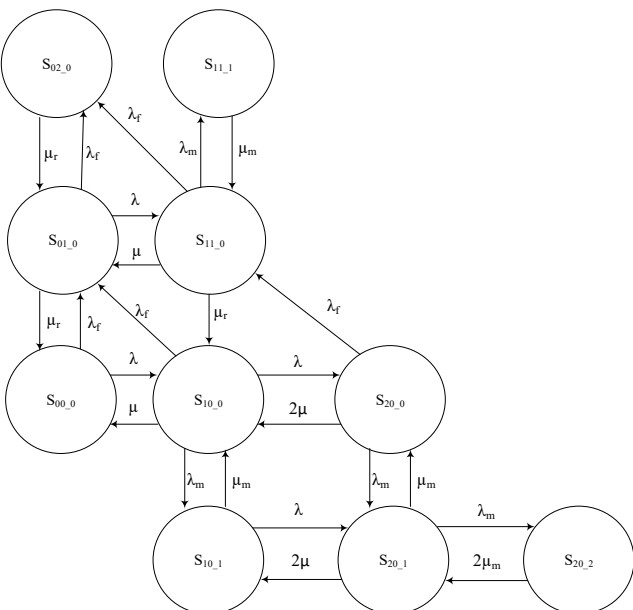

**Figure 8.** Graph of the Markov chain of the model (M/M/2):(-/0/-):(F/S).

If UAVFaaDS is described via the (M/M/2): (-/0/-):(F/S) model, the final probabilities of it being in the given states can be found by solving the following Kolmogorov–Chapman algebraic equations system:

$$\begin{cases}
-\left(\lambda + \lambda_f\right)P_{00} + \mu_r P_{01} + \mu P_{10} = 0; \\
-\left(\lambda + \lambda_f + \mu + \lambda_m\right)P_{10} + \lambda P_{00} + \mu_r P_{11} + 2\mu P_{20} + \mu_m P_{10\_1} = 0; \\
-(\lambda + \mu_m)P_{10\_1} + \lambda_m P_{10} + 2\mu P_{20\_1} = 0; \\
-\left(\lambda_f + 2\mu + \lambda_m\right)P_{20} + \lambda P_{10} + \mu_m P_{20\_1} = 0; \\
-(\lambda_m + \mu_m + 2\mu)P_{20\_1} + \lambda_m P_{20} + \lambda P_{10\_1} + 2\mu_m P_{20\_2} = 0; \\
-2\mu_m P_{20\_2} + \lambda_m P_{20\_1} = 0; \\
-\left(\lambda_f + \mu + \mu_r + \lambda_m\right)P_{11} + \lambda P_{01} + \lambda_f P_2 + \mu_m P_{11\_1} = 0; \\
-\mu_m P_{11\_1} + \lambda_m P_{11} = 0; \\
-\mu_r P_{02} + \lambda_f P_{01} + \lambda_f P_{11} = 0; \\
P_{00} + P_{10} + P_{20} + P_{01} + P_{11} + P_{02} + P_{10\_1} + P_{20\_1} + P_{20\_2} + P_{11\_1} = 1.
\end{cases} \quad (5)$$

For $\lambda = 0.5$ 1/h, $\lambda_f = 0.01$ 1/h, $\lambda_m = 0.05$ 1/h, $\mu = 0.4$ 1/h, and $\mu_r = 1$ 1/h, $\mu_m = 2$ 1/h, the final probabilities of UAVFaaDS being in the given states are as follows: $P_{00\_0} = 0.556$, $P_{10\_0} = 0.227$, $P_{20\_0} = 0.069$, $P_{10\_1} = 0.069$, $P_{20\_1} = 0.0617$, $P_{20\_2} = 2.16 \cdot 10^{-3}$, $P_{01\_0} = 6.95 \cdot 10^{-3}$, $P_{11\_0} = 2.07 \cdot 10^{-3}$, $P_{11\_1} = 5.2 \cdot 10^{-4}$, and $P_{02\_0} = 9.0 \cdot 10^{-5}$.

Thus, the probability $P_{ssd} = P_{00\_0} + P_{10\_0} + P_{20\_0} = 0.901$ and the probability $P_{is} = P_{00} = 0.556$.

Results of research of the model (M/M/2):(-/0/-):(F/S) can be presented as dependencies $P_{ssd}$ and $P_{is}$ on:

- The arrival rate (Figure 9 where $\lambda_f = 0.01$ 1/h, $\lambda_m = 0.5$ 1/h, $\mu = 1$ 1/h, $\mu_r = 1$ 1/h, $\mu_m = 2$ 1/h);
- The service rate (Figure 10 where $\lambda = 0.5$ 1/h, $\lambda_f = 0.01$ 1/h, $\lambda_m = 0.5$ 1/h, $\mu_r = 1$ 1/h, $\mu_m = 2$ 1/h);
- The rate of UAV failures (Figure 11 where $\lambda = 0.5$ 1/h, $\lambda_m = 0.5$ 1/h, $\mu = 1$ 1/h, $\mu_r = 1$ 1/h, $\mu_m = 2$ 1/h);

- The rate of UAV recovery (Figure 12 where $\lambda = 0.5 \ 1/h$, $\lambda_f = 0.01 \ 1/h$, $\lambda_m = 0.5 \ 1/h$, $\mu = 1 \ 1/h$, $\mu_m = 2 \ 1/h$);
- The rate of sending the UAVs for maintenance (Figure 13 where $\lambda = 0.5 \ 1/h$, $\lambda_f = 0.01 \ 1/h$, $\mu = 1 \ 1/h$, $\mu_r = 1 \ 1/h$, $\mu_m = 2 \ 1/h$);
- The rate of UAV maintenance (Figure 14 where $\lambda = 0.5 \ 1/h$, $\lambda_f = 0.01 \ 1/h$, $\mu = 1 \ 1/h$, $\mu_r = 1 \ 1/h$, $\mu_m = 2 \ 1/h$).

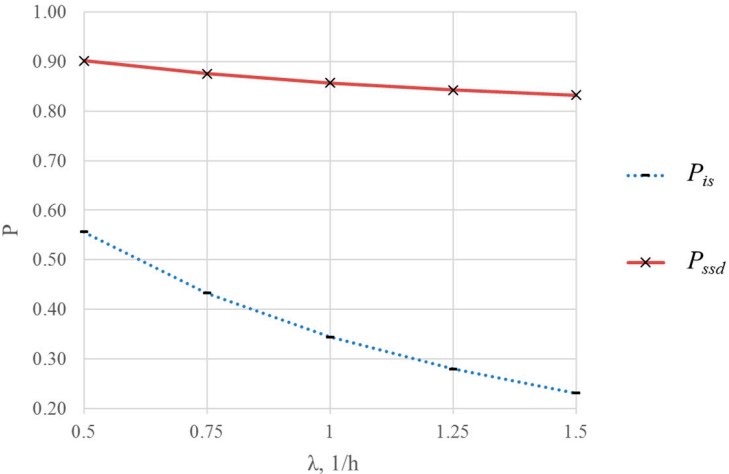

**Figure 9.** Dependencies of the probabilities of the successful service delivery $P_{ssd}$ and the idle state of the system $P_{is}$ on the arrival rate $\lambda$.

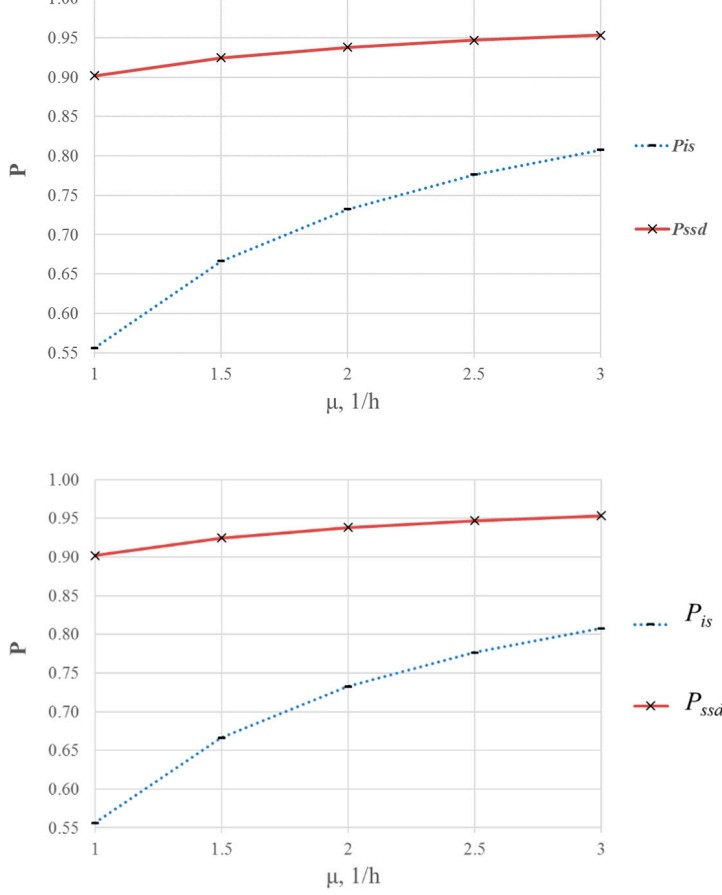

**Figure 10.** Dependencies of the probabilities of the successful service delivery $P_{ssd}$ and the idle state of the system $P_{is}$ on the service rate $\mu$.

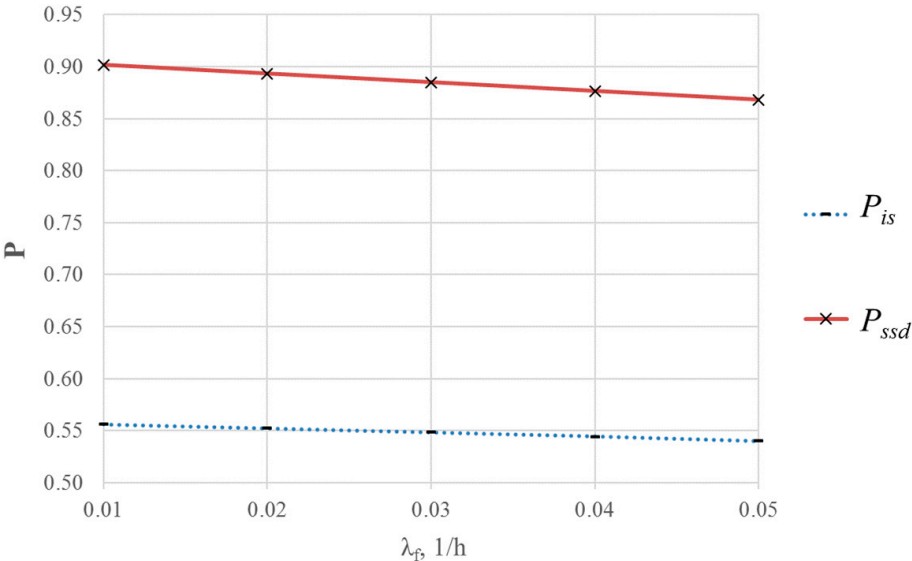

**Figure 11.** Dependencies of the probabilities of the successful service delivery $P_{ssd}$ and the idle state of the system $P_{is}$ on the rate of UAV failures $\lambda_f$.

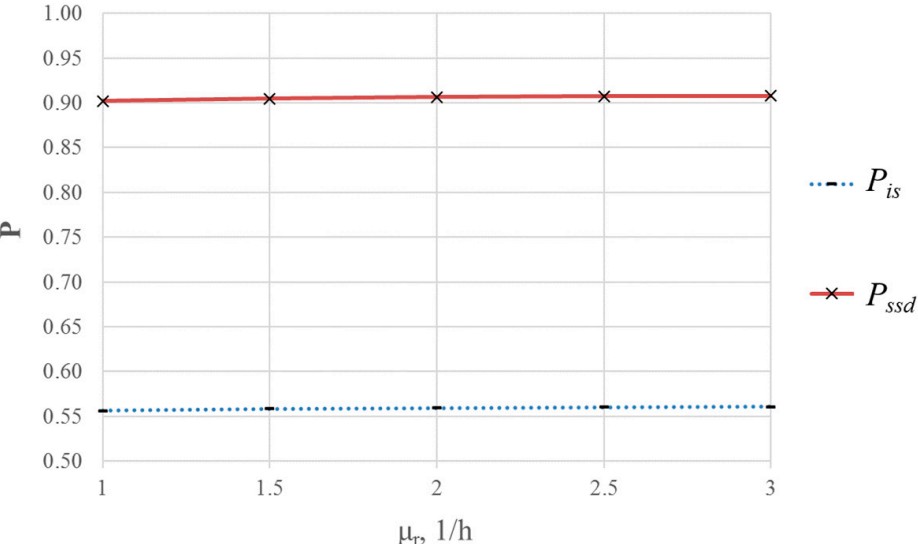

**Figure 12.** Dependencies of the probabilities of the successful service delivery $P_{ssd}$ and the idle state of the system $P_{is}$ on the rate of UAV recovery $\mu_r$.

The obtained results allow making the following conclusions:

- To increase $P_{ssd}$, it is necessary to increase the service rate, rate of UAV recovery, and maintenance rate, the increase of which also leads to the increase of $P_{is}$;
- To ensure $P_{ssd} > 0.9$, it is necessary to fulfil the condition that the value of the service rate exceeds the arrival rate by at least two times;
- An increase in UAVs' maintenance rate may require significant costs, but it leads to an insignificant increase in $P_{ssd}$.

This model can be used to analyze the functioning of UAVFaaDS considering the failures of UAVs which lead to the loss of orders, as well as the need for maintenance (recovery and charging) of UAVs during the execution of orders. The use of this model allows determining the necessary parameters, firstly, the order service flow, secondly, the recovery of UAVs, as well as for justifying the number and characteristics of UAVs and their maintenance subsystems, which are necessary for the deployment of the dependable service.

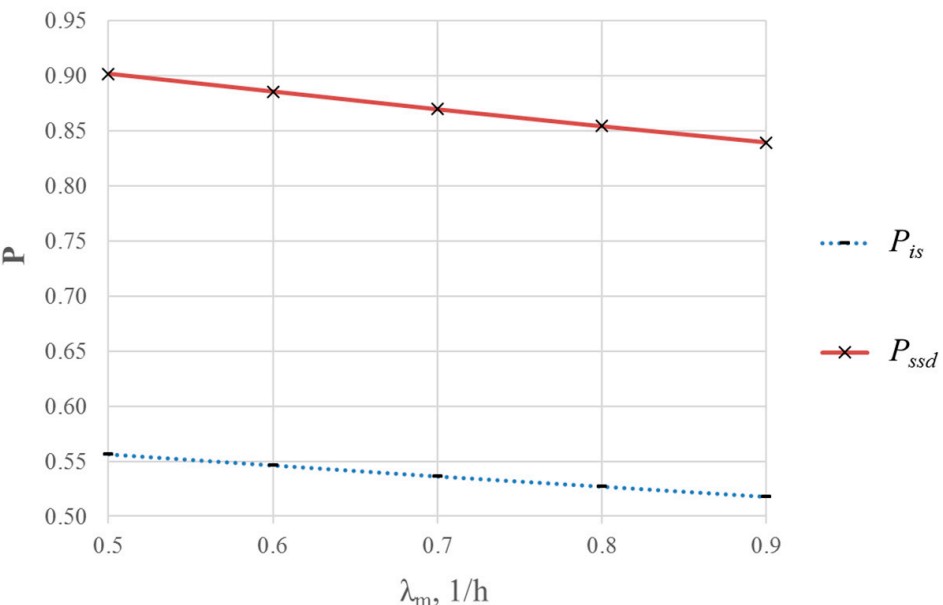

**Figure 13.** Dependencies of the probabilities of the successful service delivery $P_{ssd}$ and the idle state of the system $P_{is}$ on the rate of sending the UAVs for maintenance $\lambda_m$.

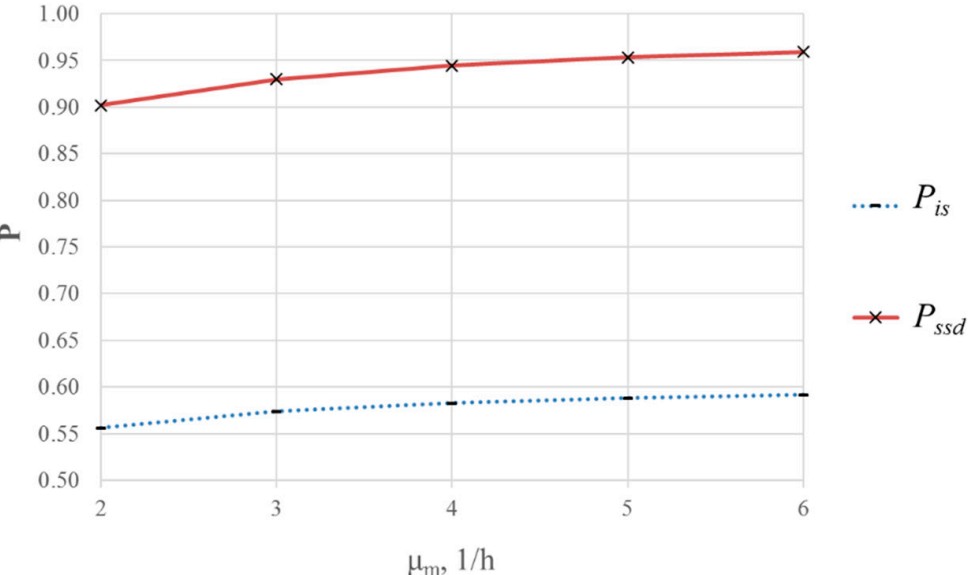

**Figure 14.** Dependencies of the probabilities of the successful service delivery $P_{ssd}$ and the idle state of the system $P_{is}$ on the rate of UAV maintenance $\mu_m$.

## 4. Case Study of UAVFaaDS Application

Let us consider two examples of UAVFaaDS applications for the delivery of medicines: in the normal mode and in the emergency mode. The organization of UAVFaaDS applications in the normal and emergency modes has a number of features and differences. The procedure for analyzing the task of drug delivery and synthesis of a system for its implementation are described in detail below. These examples are based on the experience of medicine delivery for districts of Kharkiv, Ukraine in 2022.

### 4.1. UAVFaaDS Application for Delivery of Medicines in the Normal Mode

An example of the UAVFaaDS application for delivery of medicines in the normal mode corresponds to the conditions when the entire network of pharmacies works in the normal mode and is available to city residents. The need for delivery arises in a situation

when the sick and injured, who are admitted to hospitals, need medicines that, for various reasons, are not available in the hospitals. Pharmacies located near these hospitals can be used to provide medicines in these cases.

For the efficiency of the UAVFaaDS application, it is necessary to form a hub comprising several hospitals and a hub comprising several pharmacies that utilize one fleet of UAVs located on a site near pharmacies. The arrival flow of medicine orders is formed by sending prescriptions to pharmacies by hospital departments. Pharmacies form orders as packages that are delivered by UAVs to the respective hospitals, where these packages are collected, and the UAVs return to the site where they are prepared for subsequent deliveries.

The scheme of UAVFaaDS application for delivery of medicines to the hospital hub comprising the emergency hospital, the emergency surgery institute, and the railway hospital is shown in Figure 15. Medicines are delivered from two pharmacies using different chains located nearby. The formation of a flow of the same type of applications from three hospitals determines the use of a swarm of three commercial quadcopters located at a site near two pharmacies. UAVs can fail and be recovered. Application flows and their maintenance are the simplest. The order of service is FCFS. The duration of the order servicing includes the time required for the preparation of the UAV before and after the flight. Due to the short length of UAV routes, there is no need to charge their batteries. Let us assume that there is no queue of orders.

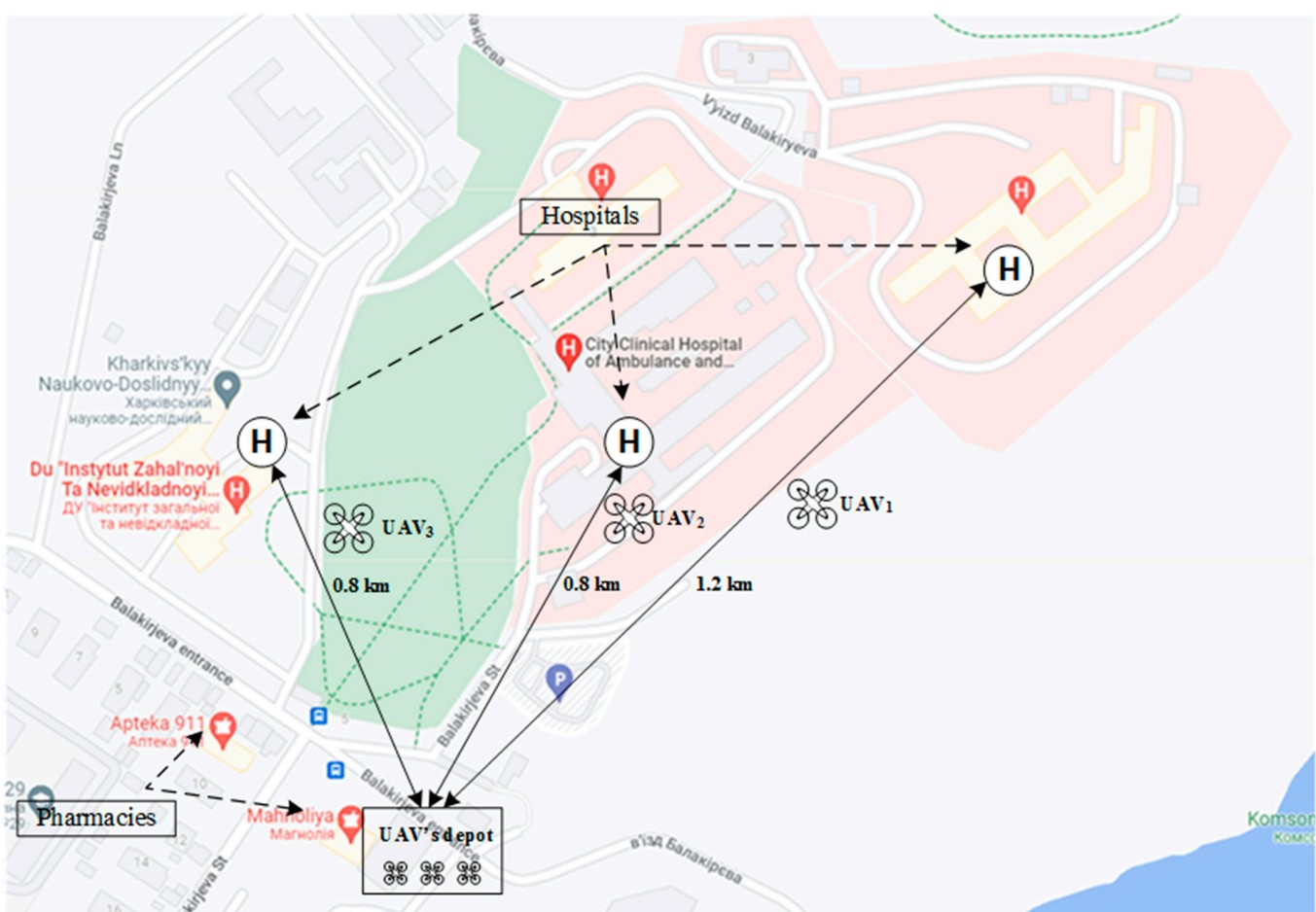

**Figure 15.** The scheme of the UAVFaaDS application for the delivery of medicines in the normal mode.

The scheme of the UAVFaaDS application for the delivery of medicines in the normal mode can be described via model (M/M/3):(-/0/-):(F).

For $\lambda = 1\ 1/h$, $\lambda_f = 0.001\ 1/h$, $\mu = 2\ 1/h$, and $\mu_r = 1\ 1/h$, the final probabilities of UAVFaaDS being in the given states are as follows: $P_{00} = 0.158$, $P_{10} = 0.316$, $P_{20} = 0.315$, $P_{30} = 0.210$, $P_{01} = 3.782 \cdot 10^{-4}$, $P_{11} = 3.305 \cdot 10^{-4}$, $P_{21} = 2.903 \cdot 10^{-4}$, $P_{02} = 2.457 \cdot 10^{-7}$, $P_{12} = 2.739 \cdot 10^{-7}$, and $P_{03} = 1.732 \cdot 10^{-10}$.

Thus, for the proposed scheme and initial data, the probability $P_{ssd} = 0.999$ and the probability $P_{is} = 0.158$.

A decrease of $\mu$ to 0.9 reduces the probability of medicine delivery without package loss to 0.94. At the same time, the probability of delivering medicines with the loss of one package is equal to 0.99.

### 4.2. UAVFaaDS Application for Delivery of Medicines in the Emergency Mode

The UAVFaaDS application for the delivery of medicines in the emergency mode differs from the normal one due to the reduced availability of the pharmacy network to city residents. The reasons for this may be restrictions on the work of pharmacies or their destruction due to emergency situations. In such cases, the distances that must be covered by UAVs to deliver medicines to hospitals located in remote areas or suburban areas increase.

For the use of UAVFaaDS in emergency mode, it is also necessary to form a hub comprising several hospitals and a hub comprising several pharmacies that utilize one fleet of UAVs located on a site not far from the pharmacies. The scheme of the UAVFaaDS application in the emergency mode is similar to the scheme in the normal one, with the difference that it is necessary to charge (replace) the UAV batteries during the service of orders due to the considerable length of the routes and the limited flight time of the UAVs.

The scheme of using UAVFaaDS for the delivery of medicines to the hub of hospitals, which includes the oncology center and the children's hospital, which are located at distances of 3.5 km and 5.5 km from the UAV site, respectively, is shown in Figure 16. Medicines are delivered from two pharmacies using different chains located nearby. The formation of a stream of similar orders from hospitals necessitates the use of a swarm of two commercial quadcopters located at a site near these pharmacies. The flow of applications and their maintenance is the simplest. The order of service is FCFS. UAVs can fail and be recovered, and the probability of UAV failures in the emergency mode is higher than in the normal one. The duration of the order processing includes the time required for the preparation of the UAV before the flight and after the flight. Due to the significant length of UAV routes, it is necessary to charge or replace their batteries while order servicing. Each UAV uses a separate ABMS. Let us assume the absolute reliability of ABMS and the absence of a queue.

The scheme of the UAVFaaDS application for the delivery of medicines in the normal mode can be described via model (M/M/3):(-/0/-):(F/S) (In Figure 8, it was already in the classification).

For $\lambda = 2\ 1/h$, $\lambda_f = 0.01\ 1/h$, $\lambda_m = 0.5\ 1/h$, $\mu = 4\ 1/h$, $\mu_r = 1\ 1/h$, and $\mu_m = 2\ 1/h$, the final probabilities of UAVFaaDS being in the given states are as follows: $P_{00\_0} = 0.555$, $P_{10\_0} = 0.277$, $P_{20\_0} = 0.069$, $P_{01\_0} = 0.006$, $P_{11\_0} = 0.002$, $P_{02\_0} = 9 \cdot 10^{-5}$, $P_{10\_1} = 0.069$, $P_{20\_1} = 0.014$, $P_{20\_2} = 0.002$, and $P_{11\_1} = 0.007$.

Thus, for the proposed scheme and initial data, the probability $P_{ssd} = 0.901$ and the probability $P_{is} = 0.554$.

The model for the emergency mode considers the need for operational maintenance and restoration of UAVs due to a possible increase in the duration of service orders. In particular, the model considers changed locations, the application of ABMSs, and considers additional parameters and their changes: the rate of sending the UAVs for maintenance and the rate of UAV maintenance.

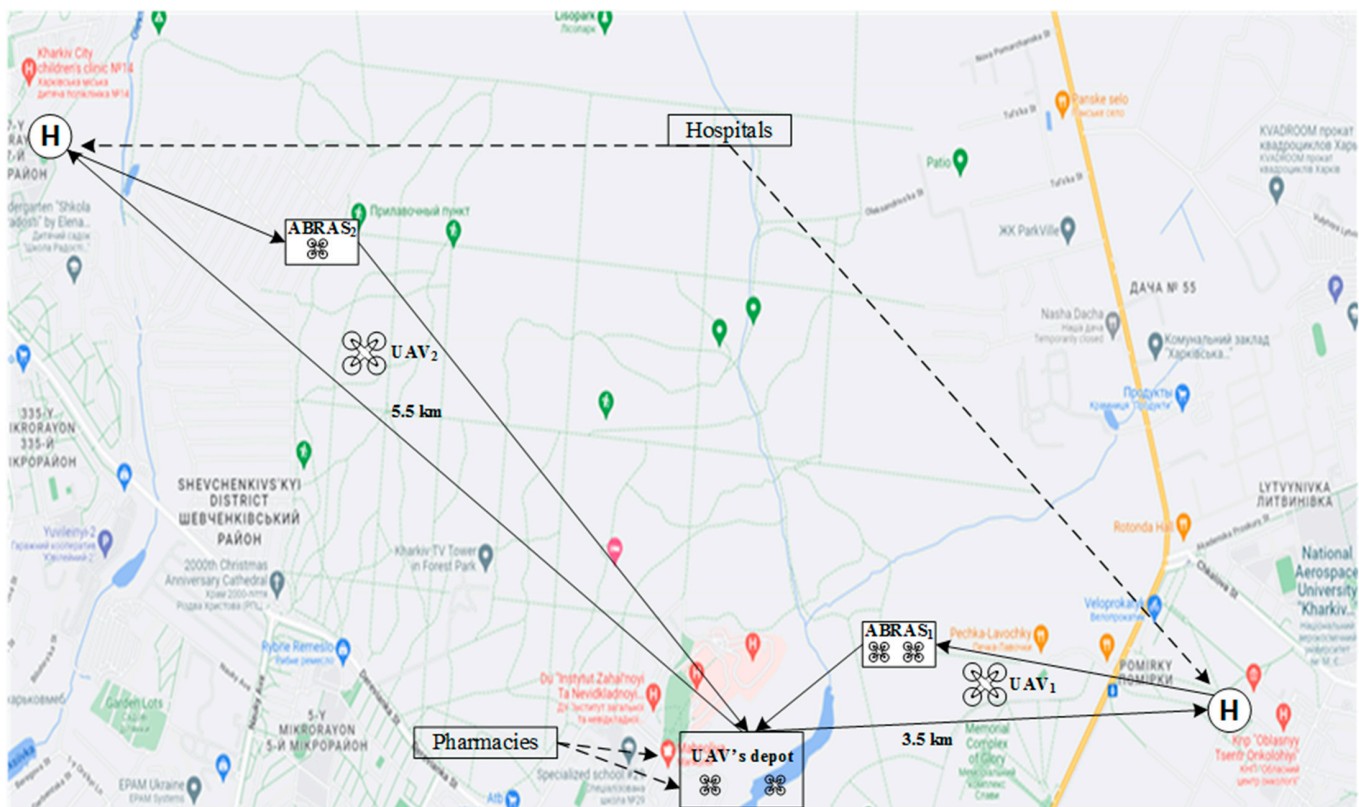

**Figure 16.** The scheme of UAVFaaDS application for the delivery of medicines in the emergency mode.

## 5. Recommendations for Using the Developed Models

The models presented in the previous sections illustrate examples of how they can be applied, firstly, to calculate the indicators of operating UAVFaaS systems and form relevant recommendations for restructuring the fleet, and secondly, to synthesize (justify the structure and parameters) the fleet and its UAVs considering the characteristics of the city and various conditions of use, in particular, normal and emergency modes. On the basis of such models, it is possible to quickly rebuild the fleet for the provision of services, in particular, the delivery of medicines when tasks and conditions change. These are the main differences between the models in comparison with studies [29,36,42]. Despite some simplification of the considered cases, they provide a certain algorithm for applying the models developed in Section 3 and confirm their correctness and practicality.

During the analysis of the UAVFaaDS deployment task, the following tasks are solved:

- Selection of models that describe UAVFaaDS in the form of queueing;
- Determination of UAVFaaDS indicators according to the specified model;
- Providing recommendations on improving UAVFaaDS indicators.

The choice of models describing UAVFaaDS in the form of queueing is made based on the proposed classification (Figure 1), moving from the left side of the classification and choosing a negative value for each classification feature. Examples of movement and selection of appropriate values of classification features during model selection are indicated with dashed lines, and the corresponding designations of the selected models are shown in the right column of Figure 1.

To determine UAVFaaDS indicators, a corresponding Markov graph of states is built according to the specified model, for which the final probabilities of the system in the corresponding states are determined. Those that characterize the PSSD and $P_{is}$ indicators are selected from the set of states and their values are determined.

In the next stage, the compliance of the system with the requirements for the provision of DS is checked. If the $P_{SSD}$ value does not meet the requirements, it is necessary to

determine ways to improve its efficiency by changing the parameters that characterize the processes of service of orders, the composition of the system, and its reliability.

During the synthesis of UAVFaaDS, the following tasks are solved:

- A detailed description of the synthesis (selection) task of UAVFaaDS;
- Selection of UAVFaaDS parameters through analysis and application of UAVFaaDS models in the form of queueing.

A detailed description of the formulation of the UAVFaaDS synthesis (selection) problem can be made using IDEF0 diagrams or verbally.

It is advisable to select UAVFaaDS parameters through iterative analysis and application of UAVFaaDS models in the form of queuing systems, moving from simple models that allow the general outline of UAVFaaDS to more complex models that specify the composition and parameters of UAVFaaDS and details permitting, considering:

(a)  operating modes, which may determine the general description of the service and the requirements for its implementation for different modes;
(b)  spatial factors, namely the size of the site areas where UAVFaaDS will be deployed;
(c)  maintenance policies determine the sequence of UAV recovery, testing, recharging, and so on during order processing.

During the synthesis of UAVFaaDS, the number and nomenclature of drones and their swarms are selected, the characteristics of which ensure the functioning of services in the specified operating modes and spatial factors, as well as the selection of the parameters of the UAV maintenance and repair system, which ensure the given level of PSSD.

## 6. Discussion

The main results of this research according to objectives are the following:

- *The concept and principles of structuring the UAV fleet as a dependable multi-service system for smart cities*. The concept of "UAV fleet as a dependable service" is based on the aggregation of multi-functional UAVs to provide various services. Productivity and dependability of services are ensured by systems of maintenance, operational backup, and recovery in case of failures and cyberattacks. This result is presented by the theoretical-set model considering sets of services, UAVs, and redundancy options. To assure the required dependability, it is needed to specify the taxonomy of failures of the UAV fleet, its information, and technical infrastructure including equipment and software of UAVs and ABMSs;
- *The dependability and performance models of UAVFaaDS and their classification*. These models are grounded on queueing theory and Markov chains considering different request streams and failures. They form a modeling base for the analysis and synthesis (composition) of the UAV fleet to cover a set of services and meet requirements for their quality. The modeling base can be extended by considering the parameters of the physical and information environment. So-called critical services should be selected into a special subset and developed in such a way as to assure a high level of dependability and safety.
- *The cases of models' application and calculating parameters and indicators of UAV fleets for delivery of medicines for normal and emergency modes*. In these cases, the parameters, capacity and space locations of components of UAVFaaDS, and the practical experience of operational staff have been taken into consideration.

The analysis of modeling results presented in Sections 3 and 4 allows concluding the following:

- To ensure a high probability of successful service delivery of UAVF, it is necessary to provide multiple excesses of the service rate over the arrival rate, which is achieved by increasing the number of UAVs and servers or by increasing their performance characteristics;
- A large value of the probability of the system idle state indicates inefficient use of the service. To reduce service downtime and provide a high probability of successful

service delivery, it is necessary to reduce the number of servers/UAVs while simultaneously improving their characteristics, which will provide the necessary service rate values;

- UAV reliability indicators do not significantly affect the value of the probability of successful service delivery for customers who need a short order processing time. With large service time values, it is necessary to consider the possibility of reducing the $P_{ssd}$ value due to the UAV failures and to use the recovery and redundancy of UAVF components;
- Exceeding the service time over the value of the time of autonomous functioning of UAVs used as servers requires the deployment of the ABMS system, which ensures the restoration of on-board power sources of UAVs. In the case of using a service system, the service rate decreases, which requires taking measures to ensure the high probability of successful service delivery by using more productive UAVs or increasing their number.

Based on the developed models and the analysis of their capabilities, a number of problems are solved, namely:

- Determination of the need to use reserve UAVs and their number to ensure the quality of service requirements ($P_{ssd}$ indicator), if the intensity of applications and their maintenance is known with a fixed intensity of failures and repair of UAVs;
- Search for the error-free characteristics of UAVs that ensure the requirements for the quality of service against known intensities of applications and their maintenance and a fixed number of UAVs;
- Determination of the number and characteristics of the ABMS reliability as a basic element of the service system, ensuring the requirements for the quality of service against known intensities of applications for the service of UAVs and their number;
- Determination of the economic feasibility of the expansion of the service (increase (expansion) of the range of UAVs and ABMSs) with an increase in the flow of orders for service. This requires the application of additional criteria considering the cost of individual UAVs and ABMSs and optimization of the value of the probability of downtime.

Concept UAVFaaDS, in fact, covers a few separate services for smart cities, dependable and resilient ecosystems [47] such as:

(a) more simple ones (these services are covered by the suggested models):

- (separate) UAV as a service;
- UAV swarm as a (dependable) service;
- UAV flock as a (dependable) service.

(b) more complex/general ones:

- UAVF as a dependable and resilient service. In this case, a few aspects (changing of requirements to UAVFaaS, parameters of the physical and cyber environment, unspecified failures, and so on [44]) should be considered. For that, combined Markov and semi-Markov models and their modifications such as multi-fragmental and multi-phase models can be applied [48];
- Autonomous transport vehicles (ATV) as a (dependable) service. Some described principles and models can be adopted and applied for other types of vehicles (driverless cars, shuttles, sea robots, and so on);
- Mixed mobile means including ATV, UAV, and others as a (dependable and resilient) service. This case covers two previous ones.

## 7. Conclusions

The main objectives of the research have been solved. The key feature of the proposed approach is describing UAVFaaDS using the set-theoretical and theory of queueing models, which allow solving the tasks of UAV groups analysis as an SCS, choice of parameters (number of main and redundant drones, attributes of their dependability, and so on), and developing structure of UAVF.

The suggested concept UAVFaaDS develops and details ideas of dependable and resilient smart ecosystems. The discussed cases have demonstrated applying elements of the UAVFaaDS concept for the delivery of medicines in normal and emergency modes.

The suggested models assess the efficiency and dependability of UAVF considering arrival requests of service delivery, failures of drones, maintenance parameters, and so on. Formally, these models do not consider reasons for failures caused by cyber-attacks on vulnerabilities of drone equipment and fleet IT infrastructure.

However, as was mentioned in Section 2, it can be considered by specifying failure rates considering physical and design faults and attacks on cyber assets. To apply the conservative approach to UAVF dependability assessment, all models presented in Section 3 can be added by a state of the cyber failure, in which transitions from all states exist. Correspondingly, reverse transitions describe recovery processes such as patching and so on. A more accurate assessment can be obtained using multi-fragmental Markov models.

In our opinion, the most important directions of future research are as follows:

- Development of the IT infrastructure of UAVFaaDS architecture joining embedded on-board systems, Internet of Drones/UAVF, $IS_C$, and $IS_M$ in accordance with Equation (1) to build a more detailed reliability block diagram or failure tree;
- Development and research of UAVFaaDS multi-fragmental and multi-phase models of availability considering detection and elimination of design faults and vulnerabilities of separate components and failures of ABMSs as well;
- Extending the queueing-theory-based set of models considering the different combinations of failures and policies of recovering drones, as well as ABMS hardware and software in case of failures;
- Specifying parameters of different SCS requests and optimization of number and types of drones, the structure of swarms and flocks for delivery of services in accordance with criteria "dependability-costs".

**Author Contributions:** Conceptualization, V.K. and A.R.; methodology, V.K. and I.K.; models, V.K., H.F. and I.K.; software, I.K.; validation, A.R. and O.I.; formal analysis, O.I.; investigation, I.K. and O.I.; resources and data curation, H.F. and I.K.; writing—original draft preparation, H.F., I.K. and O.I.; writing—review and editing, O.I. and A.R.; visualization, I.K.; supervision, V.K.; project administration, A.R.; funding acquisition, all authors. All authors have read and agreed to the published version of the manuscript.

**Funding:** This work is funded by the Ministry of Education and Science of Ukraine, project FLINT (Fundamentals and methods of dependability assurance of UAV FLeets for INTellectual systems of monitoring critical objects, No. 0121U112172, 2021–2023).

**Institutional Review Board Statement:** Not applicable.

**Data Availability Statement:** Not applicable.

**Acknowledgments:** The authors appreciate the scientific staff of the Department of Computer Systems, Networks and Cybersecurity of the National Aerospace University "KhAI" (Kharkiv, Ukraine), and Thaddeus Kochanski, Sensors Signals Systems, for invaluable inspiration, hard work, and creative analysis during the preparation of this paper.

**Conflicts of Interest:** The authors declare no conflict of interest.

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
