# Peer review of "UAV Fleet as a Dependable Service for Smart Cities: Model-Based Assessment and Application"

_smartcities, doi:10.3390/smartcities5030058_

Round 1

Reviewer 1 Report

The paper provides a good, generalized background of the topic that quickly gives the reader an appreciation of the wide range of applications for this technology.

Author Response

Dear Reviewer,

Thank your for your feedback.

Kindest regards,

Authors

Reviewer 2 Report

The following comments are needed to be covered in order to improve the overall quality:

1. Abstract needs more attention to the contribution and the numerical findings. 

2. Discussion of the proposed model needs a clearer description in sections 3 and 4.

3. Simulation results of the case study in section 4 and the associated discussion must be deep compared to previous work and models. 

4. What are the benefits of the proposed models in emergency events compared with the normal state? 

5. Revise the editing.

6. Conclusion section must summarize your outcome. No references citation in this section. 

Author Response

Dear Reviewer,

please see the attachment. All comments have been properly addressed.

Kindest regards,

Authors.

Reviewer 3 Report

I have the following concerns about the manuscript titled (UAV Fleet as a Dependable Service for Smart Cities: Model-based Assessment and Application).

1. only three keywords, I mean minimum should be 5 to 6

2.  line 59 to 77- The first letter of each bullet point should be capital

3. Low level of sentence structure- line 221- Authors write...are that. it should be that are. Line 228 to 235- again same things. Main points should be start with the first capital letter.

4. Can authors shows the graphs in colour to be looks more profound and attractive

5. Discussion should be based on discussing the main objectives in a comprehensive way and the methods which have been applied need to be discussed in detail.

6. Please study practical implications and its limitations

7. The study novelty is missing, how can the readers draw a gap when authors didn't mention the rationale of study.

Author Response

(The authors gave the same response as above.)

Round 2

Reviewer 2 Report

Authors answer all raised concerns.

Reviewer 3 Report

Good job, one round of minor spell check is required.